# Global dynamics of microbial communities emerge from local interaction rules

**Simon van Vliet**[1,2]*, **Christoph Hauert**[1,3], **Kyle Fridberg**[4], **Martin Ackermann**[5,6], **Alma Dal Co**[4,5,6,7]*

**1** Department of Zoology; University of British Columbia, Vancouver, British Columbia, Canada, **2** Biozentrum, University of Basel, Basel, Switzerland, **3** Department of Mathematics; University of British Columbia, Vancouver, British Columbia, Canada, **4** School of Engineering and Applied Sciences, Harvard University, Cambridge, Massachusetts, United States of America, **5** Department of Environmental Systems Science, ETH Zurich, Zurich, Switzerland, **6** Department of Environmental Microbiology, Eawag, Duebendorf, Switzerland, **7** Department of Computational Biology, University of Lausanne, Lausanne, Switzerland

* simon.vanvliet@unibas.ch (SvV); alma.dalco@unil.ch (ADC)

**Data Availability Statement:** The Matlab code used to solve the dynamical equations and Mathematica worksheet used to derive the analytical expressions are available on Zenodo:

## Abstract

Most microbes live in spatially structured communities (e.g., biofilms) in which they interact with their neighbors through the local exchange of diffusible molecules. To understand the functioning of these communities, it is essential to uncover how these local interactions shape community-level properties, such as the community composition, spatial arrangement, and growth rate. Here, we present a mathematical framework to derive community-level properties from the molecular mechanisms underlying the cell-cell interactions for systems consisting of two cell types. Our framework consists of two parts: a biophysical model to derive the local interaction rules (i.e. interaction range and strength) from the molecular parameters underlying the cell-cell interactions and a graph based model to derive the equilibrium properties of the community (i.e. composition, spatial arrangement, and growth rate) from these local interaction rules. Our framework shows that key molecular parameters underlying the cell-cell interactions (e.g., the uptake and leakage rates of molecules) determine community-level properties. We apply our model to mutualistic cross-feeding communities and show that spatial structure can be detrimental for these communities. Moreover, our model can qualitatively recapitulate the properties of an experimental microbial community. Our framework can be extended to a variety of systems of two interacting cell types, within and beyond the microbial world, and contributes to our understanding of how community-level properties emerge from microscopic interactions between cells.

## Author summary

Microorganisms perform essential processes on our planet. Many of these processes result from interactions between different species growing in spatially structured communities. A central goal is to understand how community processes emerge from such interactions between cells. Here we develop a mathematical framework to derive community-level properties, such as the community composition, growth rate, and spatial organization,

https://doi.org/10.5281/zenodo.5636986; the data is available at the ETH repository: https://doi.org/10.3929/ethz-b-000368486.

**Funding:** SVV was funded by the SNSF Postdoc Mobility fellowship nr. 175123, CH was funded by the Natural Sciences and Engineering Research Council of Canada (NSERC) grant nr. RGPIN-2015-05795 and RGPIN-2021-02608, MA and ADC were funded by SNSF grant nr. 31003A\_169978, Eawag, and ETH Zurich, and ADC and KF were funded by the Office of Naval Research grant nr. ONR N00014-17-1-3029. The funders had no role in study design, data collection and analysis, decision to publish, or preparation of the manuscript.

**Competing interests:** The authors have declared that no competing interests exist.

from the molecular mechanisms underlying these cell-cell interactions. We focus on mutualistic communities consisting of two cell types that need to interact with each other in order to grow. We derive equations that describe how changes in the molecular parameters of cellular interactions affect individuals' and community properties. We find that spatial structure has a negative impact on these mutualistic communities: as cells become surrounded by their own type, they have less access to the other cell type with which they need to interact to grow well. We show that our framework can also be applied to other types of microbial communities and potentially to non-microbial systems such as tissues. More generally, this work advances our understanding of how scales are connected in biological systems, both in the microbial world and beyond.

## Introduction

Biological interactions pervade all of life. Interactions at lower levels of organization can confer new functionality at higher levels. For example, interactions between different cell types determine the functioning of organs and tissues in multicellular organisms and interactions between different species determine the processes an ecosystem performs [1, 2]. In natural systems, interactions often arise in spatially structured settings, where individual entities interact mostly with others that are close by in space [3]. When interactions are local, the spatial organization of the different entities defines their network of interaction. A central question is how the properties of biological systems emerge from this network of interactions. This question has primarily been studied in the context of multicellular organisms, however it is also particularly relevant in the context of microbial communities [4, 5].

Microbial communities perform essential processes on our planet, and these processes often arise from interactions between species [6, 7]. Most microbial communities form spatially structured biofilms, where cells are embedded in an extracellular polymeric matrix that limits their movement [8]. In these communities, cells modify their local environment by secreting and taking up chemical compounds, and cells thus influence their neighbors' growth, survival, and metabolic activity [9–16]. Most interactions are mediated by diffusible molecules and their strength decays with the distance between two interacting cells [17–21]. As a result, cells only interact within a limited distance, and the spatial organization of cells within the community determines which interactions are realized.

To predict and control the functioning of microbial systems, we need to uncover how cells interact in space and how these interactions determine community-level properties, such as the community composition, spatial arrangement, and growth rate [22]. Recent studies have made progress in this direction by characterizing the spatial arrangement of cells [10, 23, 24], the range over which cells interact [10, 11, 21, 25], how the sign of the interaction affects the spatial arrangement of the community [26, 27], and how the spatial arrangement of cells affects community properties, such as their growth and species composition [5, 19, 23, 27–29].

Despite these recent advances, we do not understand well how local interaction rules scale up to determine community-level properties. In previous work, we demonstrated that local interaction rules can be measured in synthetic microbial communities [11]. There, we focused on two-dimensional cross-feeding communities, consisting of two cell types, where each type could not produce an amino acid and could only grow in mixed communities by receiving this amino acid from the other type. We showed that the two cell types interact within a small interaction range and that the growth rate of a cell increases with the fraction of the partner

type within the cell's interaction range. The interaction range and the maximum growth rate can be different for the two cell types, and describe the local interaction rules between cells.

Moreover, we previously developed a biophysical model to derive how these local interaction rules depend on the biophysical parameters of the underlying molecular exchange [11]. We found that the interaction range is mostly set by the uptake rate of the exchanged molecules, while the maximum growth rate is set by the leakage rate of these molecules. Thus, the local interaction rules between cells arise from molecular-level parameters.

Our previous biophysical model gives important insight into how local interactions rules depend on the biophysical properties of cell-cell interactions, however it has two main limitations: i) it models two-dimensional communities, while in nature most microbial communities grow in three-dimensional biofilms; ii) it does not model how the local-interaction rules affect community level properties, such as the equilibrium frequency of different cell types, their degree of spatial clustering, and the community growth rate. In this work we address these limitations in two steps. We first extend our previous model to three-dimensional communities, and we derive the local interaction rules from the biophysical parameters of the underlying cell-cell interactions. Second, we present a new graph-based model to derive the community-level properties from these local interaction rules. By combining these two models, we can directly estimate key steady-state properties of the community, such as its composition, degree of spatial clustering, and its productivity, from the biophysical parameters of the cell-cell interactions.

We apply our model to cross-feeding communities and show that community-level properties are strongly influenced by a small number of key molecular parameters underlying the cell-cell interactions (e.g., the uptake and leakage rates of molecules). Moreover, we extend the model to other types of communities and compare its predictions to data we previously obtained from an experimental cross-feeding community. Taken together, our results suggests that, at least for simple biological systems, it is possible to scale up from molecular-level properties, to individual-level properties, to community-level properties. Properties at each level can be predicted from a few key quantities of the level below (Fig 1). These findings thus further our understanding of how scales are connected in biological systems.

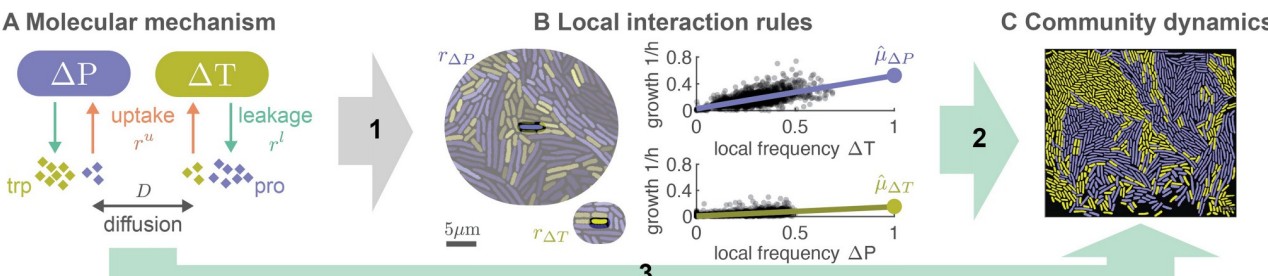

**Fig 1. A mathematical framework to scale up from molecular-level properties, to individual-level properties, to community-level properties.** We previously measured the local interaction rules for a cross-feeding community (B) and showed that these can be derived (arrow 1) from the molecular mechanisms of the interaction (A). Here we developed a mathematical model that derives community-level properties (C) either from measured local rules (arrow 2) or directly from the underlying molecular mechanisms (arrow 3). (A) The community consists of two types of *Escherichia coli*: ΔP is unable to produce the amino acid proline and ΔT is unable to produce the amino acid tryptophan. Cells exchange amino acids with the environment through active uptake (with rate $r^u$) and passive leakage (with rate $r^l$). Amino acids are exchanged between cells through diffusion in the environment (with rate $D$). All rates differ between the two amino acids. (B) Local interaction rules can be fully described by two fundamental quantities: the size of the interaction neighborhood ($r_{\Delta T}$, $r_{\Delta P}$, left); and the growth function of a cell (characterized by $\hat{\mu}$, right). Each dot corresponds to the measured growth rate of a single cell, $n = 2610$ for ΔP and $n = 2162$ for ΔT, the line shows the result of a linear regression, data reproduced from [11]. (C) We derive analytical expressions for steady state community-level properties, such as the equilibrium frequency of the two types, their spatial arrangement, and growth rate.

## Model

We present a mathematical framework that derives community-level properties from local interaction rules for a variety of systems consisting of two interacting cell types. As a case study, we focus on cross-feeding (i.e. mutualistic) interactions, however our model applies more generally to any interactions that affect the growth rate of cells, and we discuss some of these possibilities later on.

In this section we will first present a biophysical model to calculate local interaction rules as function of the biophysical parameters of the underlying cell-cell interactions (arrow 1 in Fig 1). Then we will present a graph based model to calculate steady state community-level properties as function of the local interaction rules (arrow 2 in Fig 1). Taken together, the two models allow us to calculate community level properties from the biophysical parameters of the underlying cell-cell interactions (arrow 3 in Fig 1).

### A biophysical model to calculate local interaction rules

We previously showed that two key quantities describe how cells interact in spatial cross-feeding communities. These two quantities, which we call local-interaction rules, are the interaction range (i.e. the size of the neighborhood a cell interacts with), and the maximum growth rate that cells achieves when all its neighbors are of the partner type (Fig 1B) [11]. Moreover, we developed a biophysical model to derive how these local interaction rules depend on the biophysical parameters of the underlying cell-cell interactions [11]. However, this model only applies to two-dimensional communities (such as those found in the experimental system we used), which limits its applicability to natural communities. Our first goal here is to remove this limitation, by extending the model to three dimensions.

We model a dense community of two cell-types that live in three-dimensional structures and that exchange two compounds through leakage and uptake from the extracellular environment. In S1 Text we derive analytical expression of the local interaction rules as function of the biophysical parameters of the underlying interactions. We find that the local rules in three-dimensions are identical to the ones we previously derived for two dimensions (see Fig A in S1 Text). The interaction range and the maximum growth rate are tuned independently: the interaction range primarily depends on the cell density and on the ratio between the uptake and diffusion rate of the molecules (Methods, Eq 4); the maximum growth rate primarily depends on the leakage rate of the molecules (Eq 6). Uptake, leakage, and diffusion rates are molecule specific; as a result the interaction range and maximum growth rate generally vary depending on which molecules are exchanged.

### A graph based model to describe community dynamics

The biophysical model presented in the previous section is useful to understand how local interaction rules, like the interaction range, vary with molecular parameters, such as the uptake rate of the exchanged compounds. However, the biophysical model is too complex to derive analytical expressions for community-level properties, such as as the equilibrium frequency of the two types, their spatial arrangement, and the community growth rate. We here present a graph-based model with which we can derive analytical expressions of community-level properties from the local interaction rules (arrow 2 in Fig 1). The graph-based model connects the dynamics of molecules at a lower spatial scale to the steady state properties of the communities at a higher spatial scale (arrow 3 in Fig 1). The connection relies on our ability to reduce multiple biochemical parameters to a few key combinations, namely the interaction range and the maximum growth rate, that are the real drivers of community-level properties.

Our graph-based model generalizes previous models of interacting agents based on evolutionary graph theory [30–32]. We consider a spatial system of two cell types, A and B, that exchange essential cellular building blocks (Fig 2A). We assume that the total number of cells is constant and that each cell interacts with a fixed number of neighbors. In contrast to previous models, we allow the two cell types to interact with a different number of neighbors. We

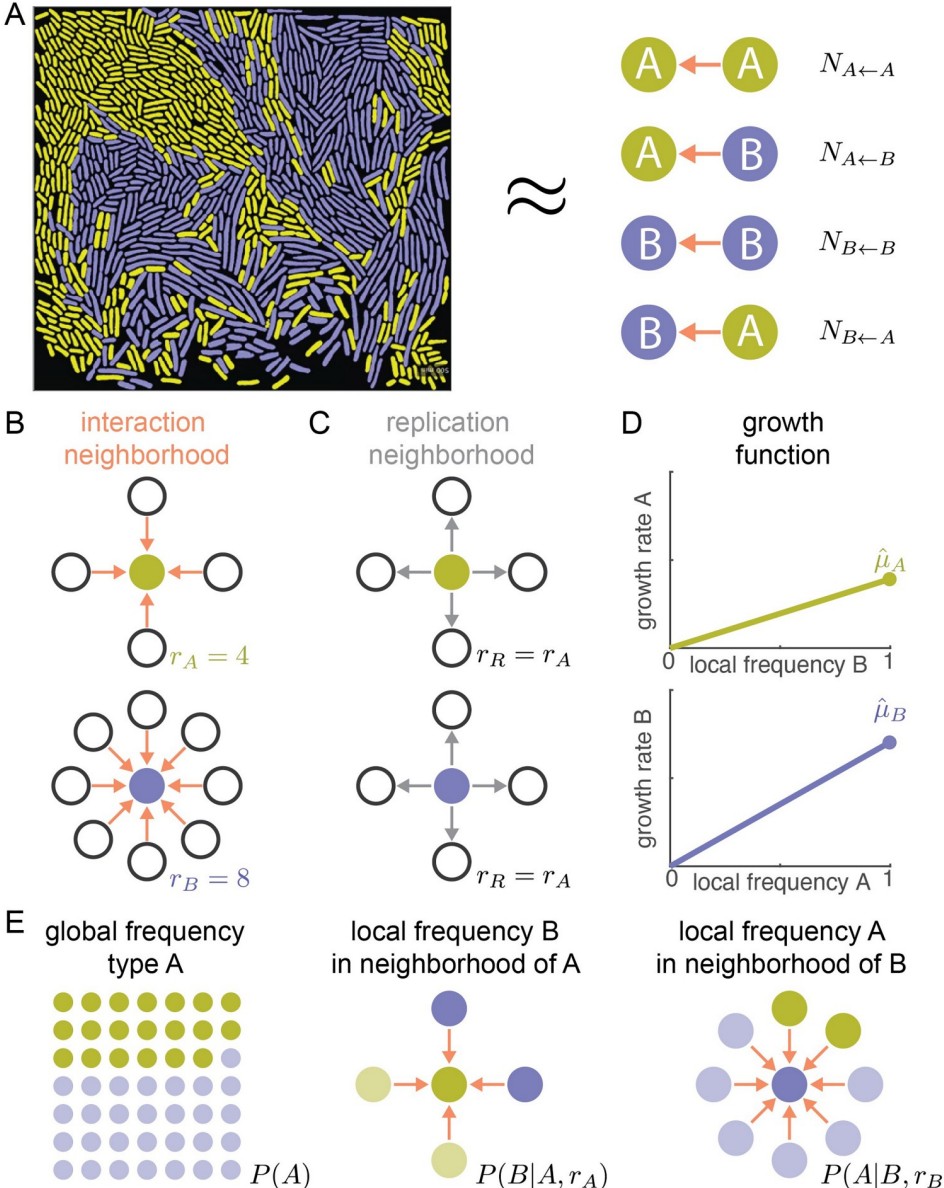

**Fig 2. Pair approximation allows for a quantitative description of spatial systems.** (A) Pair approximation assumes that a system of entities living in space and interacting with others close by (left side) can be fully described by tracking the number of all pairwise links between entities (right side). For example, a link counted in $N_{X \leftarrow Y}$ indicates that focal cell X interacts with its neighbor Y. (B-D) A system is fully described by: (B) the interaction neighborhoods of both types, characterized by the neighborhood sizes $r_A$ and $r_B$; (C) the replication neighborhood, assumed to be identical to the smallest interaction neighborhood; and (D) the growth functions of both types, characterized by the maximum growth rates $\hat{\mu}_A$ and $\hat{\mu}_B$. (E) Pair approximation has three dynamical variables that describe the global composition, $P(A)$, and local composition $P(B|A, r_A)$ and $P(A|B, r_B)$ of the system.

assume that type A interacts with $r_A$ neighbors and that type B interacts with $r_B$ neighbors. We call $r_A$ and $r_B$ the *neighborhood sizes* and arbitrarily set $r_B \geq r_A$ (Fig 2B). Which cells interact with each other is encoded in the structure of the graph: all cells that can exchange building blocks are connected by links. To allow for different neighborhood sizes, we need to use directed graphs, instead of undirected graphs. Our use of directed graphs is new in the field: previous models assumed that interactions were symmetric between cell types; some models assumed that interactions affecting birth and death rates of cells occurred at different ranges, but these ranges were the same for the different cell types [33]. Our novel combination of directed graphs with pair approximation is detailed in S2 Text.

In our model links are one-way, i.e., directed. We indicate that a cell B is linked to a cell A with B←A; this link means that the focal cell B receives building blocks from the neighbor cell A. The neighbor cell A might not have a link to the focal cell B, because cells of type A receive building blocks from a smaller distance than cells of type B. Each cell (of type A or B) interacts with all ($r_A$ or $r_B$) cells in its interaction neighborhood and these interactions determine its growth rate.

We assume that the growth rate of a cell increases linearly with the frequency of the partner type within its interaction neighborhood (Fig 2D). Each type achieves its *maximum growth rate*, $\hat{\mu}$, when it is completely surrounded by the partner type, and the maximum growth rate is different for the two types (Fig 2D). It is important to note that the *maximum growth rate* refers to the fastest growth that can be achieved when the essential building blocks are only supplied by the partner type; it is thus set by the amount of molecules that are released into the environment by the partner type. When the essential building blocks are externally provided, both cell types could potentially grow faster than they could within the cross-feeding community (i.e. their growth rate could exceed $\hat{\mu}$). In fact, we assume that both cell type can grow at the same rate $\mu^{wt}$—which is the growth rate of a non-auxotrophic wild type cell—when the building blocks are externally provided at saturating concentrations.

When a cell divides, it replaces a random neighbor within the replication neighborhood. We assume that the replication neighborhood is identical to the smaller interaction neighborhood ($r_R = r_A$, Fig 2C). This assumption has its limitations: in reality neighboring cells are not replaced, but rather pushed aside. Moreover, the replication neighborhood could be different from the smaller interaction neighborhood. However, this assumption does capture one of the most essential features of real biological systems: that cells place their offspring close by in space and thus become surrounded by their own type (see S3 Text for a more detailed discussion). Using cellular automaton simulations, we confirmed that this assumption does not strongly affect our quantitative predictions (see Fig A in S3 Text), however it makes the model analytically tractable.

We implemented our model in two complementary ways: we used a cellular automaton to simulate the dynamics numerically (see Methods) and we used pair approximation to derive analytical predictions. Overall we find that both methods agree well, as long as the communities are not too asymmetric (i.e., dominated by one of the two cell types; see Fig B in S3 Text).

Pair approximation assumes that the dynamics of a spatial system can be described by only specifying how often the different cell types are found in each others neighborhoods (Fig 2A) [30, 34, 35]. In our case, this means that the outcome of an interaction between a focal individual and one of its neighbors depends only on the identity of these two cells and not on the wider context in which these two cells are found. This approximation allows us to describe the dynamics of a spatial community (Fig 2A left) by only tracking how often cells of the different types are found within each others interaction neighborhood, i.e. by tracking the number of A←A, A←B, B←A, and B←B links (X←Y indicates that focal cell X receives building blocks

from neighbor Y, Fig 2A right). Our model therefore generalizes previous models, where interactions were assumed to be symmetric between cell types and links undirected [30].

With these assumptions, we obtain the dynamical equations for the number of pairwise links in time (see S2 Text). Because the total number of cells is constant, we only have three independent variables (e.g. the number of A←A, A←B, B←A links) and we can express the fourth one (the number of B←B links) as a function of the other three.

Instead of tracking the number of links directly, it is convenient to change variables to $P(A)$, the global probability of finding a type A cell; $P(B|A, r_A)$, the conditional probability of finding a type B cell in the interaction neighborhood of a type A focal cell; and $P(A|B, r_B)$, the conditional probability of finding a type A cell in the interaction neighborhood of a type B focal cell (Fig 2E). The first variable, $P(A)$, characterizes the global composition (i.e. the frequency of type A) of the community. The other two variables characterize the local composition of the community by specifying the average frequency of the partner type within the interaction neighborhood of a focal cell. All other probabilities can be calculated from these three variables, e.g. the global frequency of type B, $P(B)$, is given by $1 - P(A)$. Moreover, the average growth rate of a cell in a spatial system can be calculated using the average local composition, i.e. the average frequency of the partner type within the cell's interaction neighborhood.

## Results

### Steady state community properties

Pair approximation allows us to derive the global properties of the community from the local interaction rules between cells. From the neighborhood sizes ($r_A$ and $r_B$) and the maximum growth rates ($\hat{\mu}_A$ and $\hat{\mu}_B$, Fig 2B and 2D), we obtain a system of dynamical equations describing the global, $P(A)$, and local, $P(B|A, r_A)$ and $P(A|B, r_B)$, composition of the system (Fig 2E). By solving the dynamical equation for steady state, we find analytical expressions for the global and local compositions at equilibrium (see S2 Text).

The global composition of the community reaches a steady state in which the frequency of type A is given by:

$$P(A) = \frac{\hat{\mu}_A \cdot \frac{r_A - 2}{r_A} + \left(\frac{\hat{\mu}_A}{r_A} - \frac{\hat{\mu}_B}{r_B}\right)}{\hat{\mu}_A \cdot \frac{r_A - 2}{r_A} + \hat{\mu}_B \cdot \frac{r_B - 2}{r_B}} \tag{1}$$

This equation shows that the equilibrium composition of the community is mostly set by the maximum growth rates of the two types. In general, the type with highest maximum growth rate constitutes the majority (Fig 3A). Also the neighborhood size affects the composition of the community: if the interaction neighborhoods are small, the composition shifts to the type with the faster maximum growth rate (Fig 3A). Increasing the neighborhood size of even a single type moves the community composition closer to the expected composition of a well-mixed system (Fig 3B, colored lines). The neighborhood sizes do not affect the community composition when both cell types have the same maximum growth rate (Fig 3B, black line).

When the neighborhood size is large ($r_A, r_B \gg 1$), the equilibrium frequency predicted by pair-approximation (Eq 1) simplifies to the expected equilibrium frequency in a well-mixed system (see S2 Text):

$$P(A)_{WM} = \frac{\hat{\mu}_A}{\hat{\mu}_A + \hat{\mu}_B} \tag{2}$$

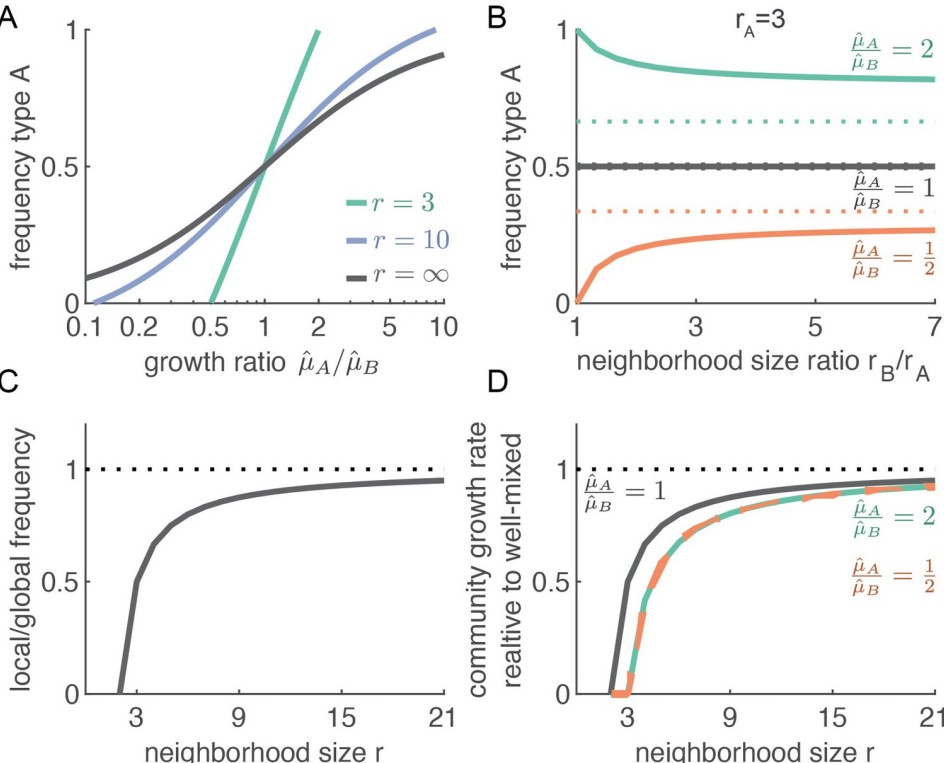

**Fig 3. Community-level properties depend on local interaction rules.** (A) The global composition of the community (global frequency of A, given by Eq 1) primarily depends on the ratio of maximum growth rates of the two types. If the neighborhood sizes of both types are large ($r_A = r_B = 10$, purple line), the equilibrium frequency approaches that of a well-mixed system (black line, given by Eq 2). If the neighborhood sizes are small ($r_A = r_B = 3$, green line), the type with the higher growth rate attains a higher frequency than in a well-mixed system. When either type becomes fixed (i.e., $P(A) = 0$ or $P(A) = 1$) cells can no longer reproduce. In natural communities, where populations sizes can change, this would lead to the collapse of the community; however, in our model we assume a constant population size, so in this case the community persists even though there is no longer any turnover of cells (i.e., community productivity drops to 0). (B) The neighborhood size affects the global composition of the community, when the two types have different maximum growth rates. The solid lines show the frequency of type A for spatial systems where $r_A$ is held constant at 3, while $r_B$ is increased. The equilibrium frequency varies as $r_B$ increases: for larger $r_B$, the frequency in a spatial system (solid lines) moves closer to the frequency in a well-mixed system (dotted lines). This result holds when the types have different maximum growth rates (red and green curves) but not when they have equal maximum growth rates (black line). (C) The neighborhood size strongly affects the local composition of the community. Here the two types have the same maximum growth rate and neighborhood size. For both types, the local frequency of the partner cells is much lower than the global frequency, when the neighborhood size is small. (D) The neighborhood size strongly affects community productivity. The community productivity (i.e. the gross production of new biomass) is smaller when cells have smaller interaction neighborhoods, because the local frequency of the partner around each cell is lower. This effect is stronger in communities where the types have different maximum growth (the red and green curves are below the black curve).

Pair-approximation is useful to understand how composition of a community changes when interaction ranges are short and neighborhood sizes small. In this regime, the equilibrium frequency given by pair-approximation (i.e. Eq 1) deviates markedly from that in well-mixed conditions (i.e. Eq 2, see Fig 3). However, when the interaction range is large, the difference in predicted equilibrium frequencies becomes largely unnoticeable (less than 1 percentage point difference when cells have more than 100 neighbors).

Pair-approximation is especially relevant if we want to investigate the local effects of spatial structure. The same global community composition can correspond to many different local spatial arrangements of the two cell types. For example, the two cell types could be highly

mixed in space or completely segregated. The spatial arrangement matters because cells interact only locally: the growth of a cell depends on the local frequency of the partner type, $P(B|A, r_A)$ and $P(A|B, r_B)$, which can be different from the global frequency, $P(B)$ and $P(A)$, when the cell types are clustered in space. A relevant quantity to describe a spatial system is thus the ratio between the local and global frequency of the partner type, and pair approximation shows that at steady state:

$$\frac{P(B|A, r_A)}{P(B)} = \frac{r_A - 2}{r_A - 1}, \quad \frac{P(A|B, r_B)}{P(A)} = \frac{r_B - 2}{r_B - 1} \qquad (3)$$

This equation shows that the frequency of the partner type in the interaction neighborhood is lower than one would expect from the global composition of the community (Fig 3C). This happens because cells place offspring close to themselves when they divide. As a consequence, the local frequency of the partner type is reduced by as much as 50% when the neighborhood size is small (e.g., when cells interact with three neighbors).

Both cell types are affected by this reduction in the local frequency of the partner type. This can be understood as follows: in a spatial system, cells of both types place their offspring close by in space and as a result they form patches. Within each patch, the frequency of the partner type is (much) lower than the global frequency. When interaction ranges are short, cells on average interact mostly within their own patch, and thus mostly interact with their own type (see Fig C in S3 Text). The average cell thus interacts with fewer cells of the other cell type than it would in a well-mixed system. In other words: in spatial systems, the local frequency of the partner type is always lower than its global frequency (as is shown by Eq 3). The difference will grow smaller as the interaction range becomes large compared to the patch size (Fig C in S3 Text). Pair-approximation shows indeed that the local frequency of the partner type approaches the global frequency when the neighborhood sizes grow large (i.e. less than 1% reduction in frequency of partner type when cells have more than 100 neighbors).

The dimensionality of the system can have a strong effect on the community properties. Given the same interaction range ($R$), cells growing in two-dimensional sheets have fewer neighbors ($\propto R^2$) compared to cells growing in three-dimensional structures ($\propto R^3$). The interaction range (i.e. the distance over which molecules are exchanged) is similar in two and three dimensions because it mostly depends on the ratio of uptake and diffusion rate (see S1 Text). However, the number of cells within the interaction range varies in two and three dimensions: cells in two-dimensional systems have fewer neighbors than cells in three-dimensional systems, which results in a stronger reduction of the local frequency of the partner type.

The reduction in the local frequency of the partner type has an important consequence for the community: cells have less access to the resources they need for growth and as a result birth rates go down. To quantify this effect, we calculate the average birth rate of cells in the community; this quantity corresponds to the gross rate at which new biomass is produced and we refer to it as the *community productivity*. It is important to note that the net change in biomass (i.e., the change in population size) also depends on the rate of cell death/loss. In our model we assume a constant population size, so the net rate of biomass production is always zero. Biologically, this corresponds to biofilms where cell growth and loss balance each other, e.g., because cells at the edge of the biofilm are flushed away. Even in such scenarios where population sizes are constant, community productivity is still an important quantity as it measures the turn-over rate of cells, which in turn affects many relevant quantities such as the overall metabolic rate in the community. We can derive an analytical expression for the community productivity and compare it to the productivity of an equivalent community growing in well-mixed condition (see S2 Text).

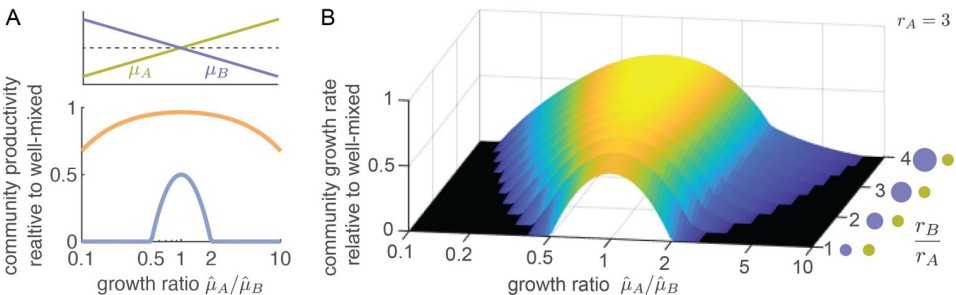

**Fig 4. Small interaction neighborhoods can lead to community collapse.** (A) When the neighborhood size is small ($r_A = r_B = 3$, purple line), asymmetric communities, where the cell types have different maximum growth rates, have low productivity. When the asymmetry is too large, communities cannot grow (i.e. community productivity is 0) in a spatially structured environment even though they could grow in well-mixed environments. When the neighborhood size is large ($r_A = r_B = 30$, orange line), the productivity of the spatial community is close to that of the well-mixed community even when there is an asymmetry. (B) Increasing the neighborhood size of just one the types is enough to increase the productivity of spatial communities and prevent collapse. $r_A$ is held constant at 3, while $r_B$ is increased.

For cross-feeding communities the productivity in spatial systems is always lower than that in well-mixed systems and this difference increases as interaction neighborhoods become smaller (Fig 3D). This reduction in productivity is more pronounced when the two cell types have different maximum growth rates. We call such communities *asymmetric*. Our model shows that the more asymmetric a cross-feeding community is, the more the community's productivity is hindered by small neighborhood sizes, to the point that the community collapses when neighborhood sizes are very small (Fig 4A). Increasing the neighborhood size of either one (Fig 4B), or both cell types (Fig 4A), can prevent the collapse of the community. This finding shows that there is a limit to the stability of cross-feeding interactions in spatially structured communities: cross-feeding cell types with different maximum growth rates that can stably coexist in well-mixed environments might not be able to survive in spatially structured environments. For cross-feeding communities, spatial structure can thus have a detrimental effect.

## Small interaction neighborhoods affect other types of communities

In this section, we will generalize our model and relax certain assumptions. Our model assumed that uptake and leakage rates differ between chemical compounds, but not between cell types. Our first goal is to relax this assumption. This assumption typically holds only for microbial communities that consist of related strains and species, or for different cell types in a multicellular organism. In these cases all cell types likely share the same uptake and leakage pathways. However, uptake and leakage rates could differ both between chemical compounds and between cell types, e.g. because cell types use different uptake transporters or differ in their membrane permeabilities. Also in this general scenario we can derive expressions for the local rules, i.e. maximum growth rate and neighborhood size, as function of the molecular parameters of the molecular exchange (see Methods Eqs 7 and 8 and S1 Text). With these local rules, we can derive analytical predictions for the community-level dynamics of any two species cross-feeding community.

So far, we have focused on cross-feeding systems were the growth rate of each cell type increases with the frequency of the other type within the interaction neighborhood. However, the growth rate of cells could depend on the partner type differently and our framework can also be applied to these systems. We will illustrate this using two examples of cells interacting by non cross-feeding interactions. First, we consider a system where a type's growth rate

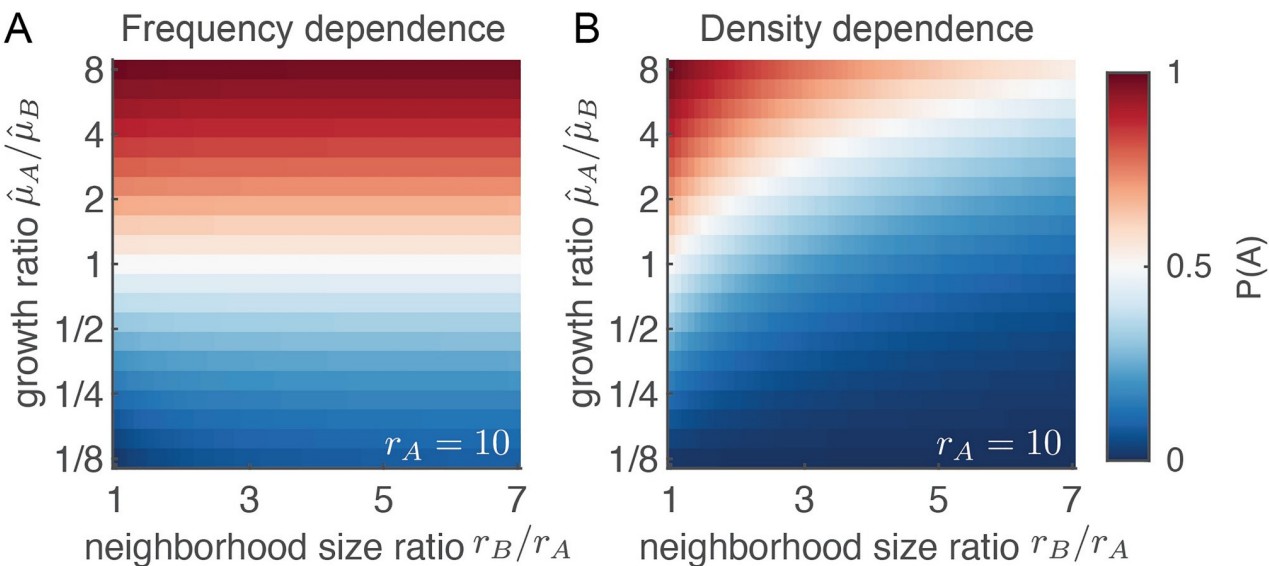

**Fig 5. The neighborhood size strongly affects the equilibrium frequency when the growth rate of a cell depends on the absolute number of partner cells.** (A) When the growth rate depends on the *frequency* of the partner type within the interaction neighborhood (frequency dependence), the equilibrium frequency of type A is almost completely determined by the ratio of the maximum growth rates (Eq 1). (B) When the growth rate depends on the *number of cells* of the partner type within the interaction neighborhood (density dependence), the equilibrium frequency of type A depends both on the ratio of the maximum growth rates and on the ratio of the neighborhood sizes (see Equation 32 in S3 Text). The slow growing type can still dominate the community when it has a much larger interaction range (blue region, top right). In both panels $r_A = 10$.

increases with the number, rather than the frequency, of partner type within the interaction neighborhood (Fig 5). This is the case when cells exchange growth-affecting molecules that are not taken up by the cells, but that are only sensed. If such molecules degrade, then the interaction range between cells can still be short even in the absence of uptake. Such molecules could for example be certain growth factors, which are rapidly inactivated by enzymes in the extracellular matrix that surround cells [36]. In this case, the type with the highest product of $\hat{\mu} \cdot r$ reaches the highest global frequency (see S2 Text). Therefore, the global composition of the community depends strongly both on the maximum growth rate and on the neighborhood size, rather than depending only on the maximum growth rate (Fig 5).

Second, we consider systems where two cell types inhibit each other's growth. In such systems, a cell's growth rate decreases (linearly) with the frequency of the other type within the interaction neighborhood. We find that cells that inhibit each other grow slower when they are well-mixed compared to when they interact within a small neighborhood (see S2 Text). This happens because cells typically are surrounded by their own type, and thus have reduced interactions with the growth inhibiting cells.

In general, our framework can be applied to any system of two cell types that affect each others growth by interacting within a finite range. If the growth rate of a cell depends linearly on the composition of its neighborhood, we can find an analytical solution for the steady state community properties; for non-linear functions, the model can be solved numerically (see S2 Text). The global composition of the community at steady state ($P(A)$) depends strongly on the chosen growth functions. However, the relation between the local and global composition of the community is always the same: i.e. for all growth functions we find that Eq 3 holds (see S2 Text).

## Comparing model to experimental microbial community

We tested our mathematical framework by comparing its predictions to experimental observations we previously obtained for a two-dimensional cross-feeding community [11]. This community consists of two cell types: one type unable to produce proline ($\Delta P$), and one type unable to produce tryptophan ($\Delta T$). The two cell types were grown together in shallow microfluidic chambers in which cells grow as two-dimensional monolayers. The chambers open on one side into a deeper flow-channel. As cells grow, they push each other into the flow-channel thereby keeping the population size inside the chambers approximately constant. We followed the dynamics inside 22 replicate chambers using time-lapse microscopy and used an automated image analysis workflow to quantify the composition, spatial arrangement, and growth rate of the community over time. These chambers are identical to each other, except for the initial frequency of the two cell types, which was determined by chance during the stochastic cell-loading process (see reference [11] for details).

To predict the properties of our experimental community, we can parameterize our model in two ways (S2 Table): we can use the biophysical parameters describing the amino acid exchange (arrow 3 in Fig 1) to calculate the neighborhood size ($r$, Eq 4), and the maximum growth rate of a cell ($\hat{\mu}$, Eq 6). Or, alternatively, we can parameterize the model using the measured interaction range and measured maximum growth rate (arrow 2 in Fig 1).

The main advantage of our framework is that it is possible to predict community level properties directly from literature values of the underlying biophysical rates; to demonstrate this possibility we present here the results based on this parameterization. The alternative parameterization, directly from the measured local rules, gives very similar results (S3 Table).

Our model recovers the experimentally observed equilibrium composition of the community. In the experiments, the frequency of the $\Delta T$ type converges over time to a stable frequency across 22 communities (Fig 6A). Our model predicts similar dynamics: both pair approximation (Fig 6B) and simulations (Fig 6C) show that the community reaches a stable equilibrium. We did not set out to accurately model the temporal dynamics of the community before it reaches steady-state; nonetheless both in the data and in pair-approximation we observed that this process takes on the order of tens of hours, though the observed dynamics are slower by about a factor 2 (Fig 6A and 6B). A more detailed investigation is needed to assess the extent to which pair-approximation can be used for non-equilibrium dynamics, but we leave this for future work. From Eq 1 we find a predicted equilibrium frequency of $\Delta T$ of 0.20, which matches well with both the experimentally observed value of 0.19 (95% CI: 0.17–0.20) and cellular automaton simulations value of 0.18 (95% CI: 0.18–0.19; Fig 6D, S3 Table). In fact, there is no significant difference between the experimentally measured equilibrium frequency, and those predicted by pair-approximation or the cellular automaton simulations ($p = 0.05$ and $p = 0.52$, respectively, t-test). It is important to note, however, that all these frequencies are close the equilibrium frequency of 0.22 expected in a well-mixed system. This is largely due to the fact that the neighborhood size of $\Delta P$ is rather large ($r_{\Delta P} = 130$); as a result, a non-spatial model can predict the global composition of our community relatively well, though there is a significant difference between the predicted and observed equilibrium frequency ($p = 3 \cdot 10^{-4}$, t-test). However, a spatial model is needed to predict other community level properties, such as the local arrangement of cells, and the productivity of the community.

An important prediction of our model is that cells becomes surrounded by their own type, reducing the local partner frequency which in turn reduces the community productivity (Fig 3C and 3D). This is also what we observe in our experimental data: for both $\Delta P$ and $\Delta T$, the local frequency of the partner type is reduced compared to the global frequency (Fig 6E). Moreover, we previously showed that this spatial clustering leads to a reduction in community

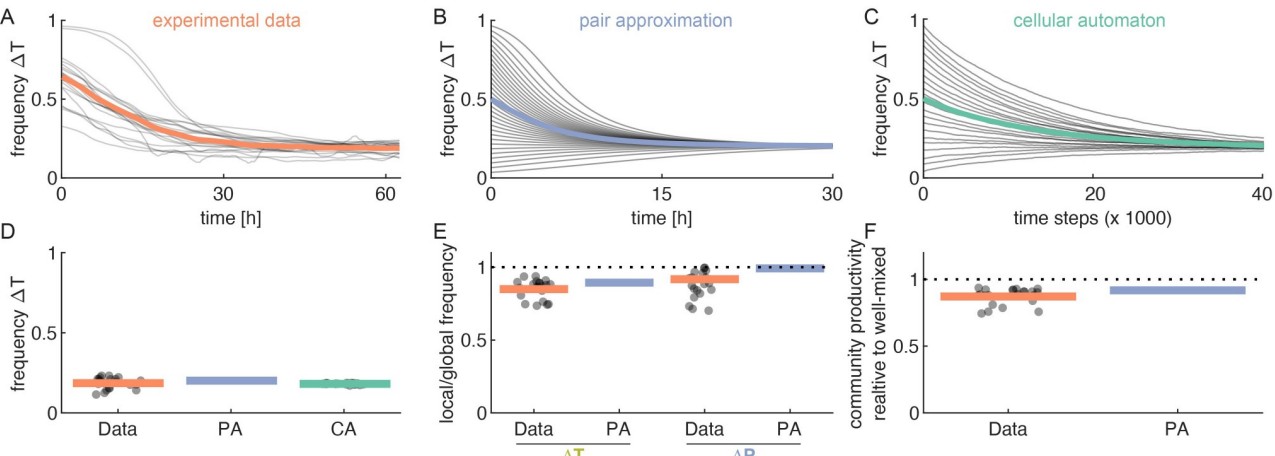

**Fig 6. Our mathematical framework can explain experimentally observed community properties.** The model parameters for the experimental community were calculated from the biophysical model (S2 Table). (A) Experimental communities approach a stable frequency of $\Delta T$. Individual communities (thin lines, n = 22) and their average value (thick line) are shown. The initial frequency of $\Delta T$ in the communities is determined by the initial number of cells that enter the microfluidic growth chambers and the subsequent growth before the start of image acquisition; as a result, it cannot be controlled experimentally. (B) Pair approximation predicts a unique stable equilibrium. The dynamical equations (Equations 12–14 in S2 Text) were solved numerically starting from different initial frequencies. (C) Cellular automaton simulations also reach a unique stable equilibrium. (D) The observed equilibrium frequency of $\Delta T$ is consistent with the model predictions. Data: $P(A) = 0.19$ (95% confidence interval (CI): 0.17–0.20), Pair-approximation: $P(A) = 0.20$, and cellular automaton: $P(A) = 0.18$ (CI: 0.18–0.19, evaluated after 100'000 time steps). The difference between the model prediction and data is less than 8%. (E) The frequency of the partner cell type within the interaction neighborhood (local frequency) is lower that the global frequency because cells are surrounded by their own offspring. Dots show measurements for 21 replicate communities, bar shows mean value. Pair approximation predicts (Eq 3) a decrease in frequency of 0.99 for $\Delta P$ and 0.89 for $\Delta T$, the experimental values are 0.92 (CI: 0.84–0.99) and 0.85 (CI: 0.82–0.88), respectively. The difference between the model prediction and data is less than 13%. (F) The average growth rate of the community is reduced due to cell clustering. An in-silico analysis (see Methods) shows that the growth rate in clustered communities, with experimentally observed spatial arrangements, is reduced by a factor of 0.87 (CI: 0.85–0.90) compared to randomized communities, where cell clusters were disrupted (Data reproduced from [11]). Pair approximation predicts a decrease by a factor of 0.92 (S3 Table); the difference between the model prediction and data is less than 6%.

productivity (Fig 6F) [11]. Qualitatively our model can explain both these effects, however the model underestimates the reduction in local partner frequency and community productivity by about a factor of 2 (Fig 6E and 6F and S3 Table). This is likely due to the simplified representation of cells and space in the model, which is needed for the sake of analytical tractability. These experiments thus suggest that the primary power of our model is to qualitatively explain how (changes in) biophysical parameters effect community level properties.

## Discussion

We developed a mathematical framework to describe the community level properties of spatially structured communities of two species from information about the cell-cell interactions that take place (Fig 1). We primarily focused on microbial cross-feeding systems and used pair approximation to derive analytical expressions for the community-level properties from knowledge of the local interaction rules (Fig 2). These rules are defined by two fundamental quantities: the size of the interaction neighborhood and the maximum growth rate that cells achieve when they are completely surrounded by the partner type; both quantities typically differ for the two cell types in the community.

We showed that these local interaction rules can directly be derived from key biophysical parameters of the underlying molecular mechanisms, such as uptake and leakage of the exchanged molecules (Fig 1). We worked out expression for the local interaction rules from the biophysical parameters for several scenarios demonstrating how our framework can help

elucidate how local interaction rules arise from the molecular exchanges, and how the local interaction rules scale up to determine community properties. For spatial cross-feeding communities, we found that the local and global properties can change independently. The global composition of the community (i.e. the equilibrium frequency of the two types) is set by the ratio of the maximum growth rates, which mostly depends on the leakage rates of the exchanged metabolites. In contrast, the composition of the local neighborhood, the one that matters the most for the cells, is set by the neighborhood size, which mostly depends on the ratio of the uptake and diffusion rates. Different biophysical parameters thus control the global and the local properties of the community. Generally, the framework we developed allows to scale up from molecular-level properties, to individual-level properties, to community-level properties.

Small neighborhood sizes reduce the frequency of the partner type around each cell. This happens because cells place their offspring close by in space. In our model we assumed that cells cannot actively move. However, cells could overcome the negative effect of having a small neighborhood size if they could actively move to locations with a higher frequency of the partner type. Without such active movement, cross-feeding cells in a spatial system have a lower birth rate than in equivalent well-mixed system (Fig 4), reducing the overall productivity of the community. This can even lead to the collapse of the community: cells in asymmetric communities (i.e. communities where the two type have different maximum growth rates) can stably coexist in well-mixed system, yet they might not coexist in a spatial system, if they interact at small ranges (Fig 4). Moreover, dimensionality matters: given a fixed interaction range ($R$, determined by a set of molecular parameters) cells growing in two dimensional colonies typically have fewer neighbors ($r \propto R^2$) compared to cells growing in three dimensional structures ($r \propto R^3$). Communities will thus have a lower productivity in two-dimensional colonies than in three-dimensional clusters, provided that the interaction range is largely independent from the dimensionality of the system.

Our biophysical model suggests that the interaction range is indeed comparable between two- and three-dimensional systems (Fig A in S1 Text). However, a recent study came to the opposite conclusion: van Tatenhove-Pel *et al.* (2020) [21] found that the interaction range in three dimensions is shorter than in two dimensions. This difference can likely be explained by the different spatial arrangement that were considered: we calculated the interaction range around larger patches of producer cells, while van Tatenhove-Pel *et al.* (2020) [21] calculated it for a single, isolated, producer cell. When there is only a single producer cells, only few molecules are produced and increasing dimensionality decreases the interaction range as they are spread over a larger volume. However, when the number of producer cells is larger (as will be the case in most communities that are not too asymmetric) this effect becomes less important and the interaction range becomes largely independent of dimensionality, as we found. Despite these differences, both our work and that of van Tatenhove-Pel *et al.* (2020) [21] show clearly that spatial structure and dimensionality have important effects on the dynamics of microbial communities.

We tested our framework using an experimental community of two cross-feeding bacteria. We found that we can quantitatively predict the equilibrium composition of our experimental community, however a well-mixed model could do so as well (Fig 6D). For the particular community we studied, spatial structure thus appears to have a negligible effect on the global composition of the community. Despite this, spatial structure has important consequences for this community: it causes cells to become surrounded by their own type, which leads to a reduction in community productivity (Fig 6E and 6F). These effects cannot be predicted by a well-mixed model, but they are explained, at least qualitatively, by our pair-approximation model. Quantitatively our model performed less well in predicting these properties, however it can still give a

order-of-magnitude estimate of the extend to which spatial structure matters for a given community.

To properly validate our model, it would be essential to perform additional experiments with different communities. This could be done in several ways: biophysical parameters (and thus local rules) vary between molecules, species, and transport pathways; changing any of these would place a community in a different part of the model parameter space. For example, one could repeat our experiments using communities that exchange different building blocks or that are composed of different species. Moreover, one could directly change relevant biophysical parameters using genetic engineering, for example by tuning the uptake rate of molecules by manipulating the corresponding uptake pathways. Finally, thanks to recent advances in image analysis techniques [37], it has now become feasible to perform similar experiments in three-dimensional biofilms, making it possible to experimentally test the effects of dimensionality on community-level properties.

Previous work has shown that short-range interactions provide an advantage to cooperative interactions, because they separate cooperative types from non-cooperators [28, 38–40]. In general, when it is good to be surrounded by your own type, we expect short-range interactions to be beneficial, as for example when two cell types inhibit each other. When it is good to be surrounded by the other type, we expect short-range interactions to be detrimental, as for example when two cell types exchange beneficial resources. When a community consists of more than two cell types, the situation can be more complex. Previous studies have shown that short-range interaction can be beneficial for cross-feeding communities that contain non-producing cells: the short-range interactions harm the producing cells, but they harm the non-producers even more, and thus prevent them from taking over [23, 41, 42].

Our framework can also model systems beyond the bacterial world, as long as they are composed of two interacting types. For example, a central question in tissues homeostasis is how two (or more) different cells types can control each other's growth by exchanging diffusible growth factors to maintain proper tissue functioning [43, 44]. These cellular systems are typically spatial, in the sense that interactions act on a finite distance. From the point of view of modeling, spatial structure introduces complexity in the mathematical representations, as well as increasing the parameter space of models. Our framework provides a simple approach to study the equilibrium properties and can assist in the design of synthetic systems and tissue engineering, as it allows for the prediction of system-level properties from molecular scale parameters [45].

Overall, our model offers a versatile representation of spatial system of interacting cells that can be adapted to various types of interactions. Many biological systems are spatially structured and multi-scale: they consist of individual entities (e.g. cell types or species) that interact with each other in space. Interactions at different levels determine the global properties of the system (e.g. multicellular organism or microbial community). To understand such biological systems, it is important to scale between levels of organization. Our work provides a contribution to this effort by creating a mathematical framework that can scale from molecular mechanisms, to local interaction rules, to global system properties.

## Methods

### Model assumptions

- Two cell types, A and B, fully occupy a regular directed graph. Type A interacts with $r_A$ neighbors, type B interact with $r_B$ neighbors, where $r_B \geq r_A$.

- A cell's growth rate increases linearly with the frequency of the other type within its interaction neighborhood.

- Individuals reproduce with a probability proportional to their growth rate, and their offspring replaces a random neighbor within the replication neighborhood.

- The replication neighborhood is identical to the smallest interaction neighborhood $r_R = r_A$.

We used pair approximation to derive analytical predictions and a cellular automaton to run simulations of the model.

## Pair approximation

We track the number of all pairwise links $N_{X \leftarrow Y}$, where $X, Y \in A, B$. The total number of links changes through time, because the neighborhood sizes are different for the two cell types (see S2 Text). This is in contrast to previous work where the two neighborhood sizes are the same and the total number of links remains constant. Two events can change the number of links: a type A cell reproduces and replaces a B neighbor, with rate $T^+$, or a type B cell reproduces and replaces an A neighbor, with rate $T^-$. During a $T^+$ event the number of A cells thus increases by one, and during a $T^-$ event it decreases by one. The rate $T^+$ is given by:

$$T^+ = P(A) \cdot P(B|A, r_A) \cdot \frac{1 + P(B|A, r_A)(r_A - 1)}{r_A} \hat{\mu}_A$$

where the first factor gives the probability of choosing a type A focal cell, the second the probability that this A cells has a type B neighbor, and the third the average growth rate of a type A cell that has at least 1 type B neighbor.

Similarly, for $T^-$ we have:

$$T^- = P(B) \cdot P(A|B, r_B) \cdot \frac{1 + P(A|B, r_B)(r_B - 1)}{r_B} \hat{\mu}_B \cdot \frac{P(A|B, r_A)}{P(A|B, r_B)}$$

The extra factor at the end is the probability that a type A neighbor that is in the (larger) interaction neighborhood is also part of the smaller replication neighborhood. We can express all probabilities as function of the number of links:

$$P(A) = \frac{N_{A \leftarrow A} + N_{A \leftarrow B}}{rA \cdot N}, \quad P(B|A, r_A) = \frac{N_{A \leftarrow B}}{N_{A \leftarrow B} + N_{A \leftarrow A}}$$

$$P(B) = \frac{N_{B \leftarrow A} + N_{B \leftarrow B}}{rB \cdot N}, \quad P(A|B, r_B) = \frac{N_{B \leftarrow A}}{N_{B \leftarrow A} + N_{B \leftarrow B}}$$

where $N$ is the total number of cells in the systems.

When a $T^+$ or $T^-$ event happens, the number of $X \leftarrow Y$ link changes by $\Delta^+_{XY}$ and $\Delta^-_{XY}$, respectively. These changes can be fully expressed as function of $N_{X \leftarrow Y}$, as we show in S2 Text. We can then write the dynamical equation for $N_{X \leftarrow Y}$ as:

$$\frac{dN_{X \leftarrow Y}}{dt} = T^+ \cdot \Delta^+_{XY} + T^- \cdot \Delta^-_{XY}$$

We can solve these equations numerically to obtain the temporal dynamics, and we can solve them analytically for the steady state, see S2 Text for full derivation.

## Model parameterization

To parameterize the model, we need to specify the size of the interaction neighborhood $r$ and the maximum growth rate of a cell $\hat{\mu}$. We previously showed that that interaction-range ($R$) of

a cell (the distance over which amino-acids can be exchanged) is given by [11]:

$$R = \beta \sqrt{\frac{2(1-\rho)^2 \cdot D}{\rho(2+\rho) \cdot (r^u + r^l)}} \ln\left[\frac{r^l}{\gamma}\left(1 + \sqrt{1 + \frac{4\gamma}{r^l}}\right) + 4\right] \tag{4}$$

where $\beta$ and $\gamma$ are constants, $\rho$ is the volume fraction occupied by cells, and $D$, $r^u$, and $r^l$ are the rates of diffusion, uptake, and leakage of the exchanged metabolite (see S1 Text for details and S1 Table for parameter values). The number of neighbors in 2D can be directly calculated from the interaction range:

$$r = \left(2R(l-w) + \pi\left(2R + \frac{w}{2}\right)^2 - \pi\left(\frac{w}{2}\right)^2\right) \cdot \rho_{2D} \tag{5}$$

where $l$ and $w$ are the average length and width of cells, and $\rho_{2D}$ is the number of cells per area. A similar expression can be derived for the number of neighbors in a three-dimensional system (see Equation 22 in S1 Text).

We previously showed that that maximum growth rate of a cell $\hat{\mu}$ is given by [11]:

$$\hat{\mu} \approx \mu_{wt} \cdot \frac{r^l}{\gamma}\left(\sqrt{1 + \frac{2\gamma}{r^l}} - 1\right) \tag{6}$$

## Generalized model parameterization

In the previous section we assumed that the uptake and leakage rates differ between molecules, but not between cell types. Here we relax this assumption and show the local interaction rules for systems where rates differ both between molecules and cell types. In S1 Text we show that in this case the maximum growth rate is given by:

$$\mu^{max} \approx \mu_n \cdot \theta\left(\sqrt{1 + \frac{2}{\theta}} - 1\right)$$

$$\theta = \frac{r_p^l I_p^C}{2\mu_n K_n} \cdot \frac{r_n^u + r_n^l}{r_p^u + r_p^l} \tag{7}$$

Here, the subscript $n$ refers to parameters of non-producing cells, while $p$ refers to parameters of the producing cells. $I_p^C$ is the internal concentration of the produced molecule inside producer cells, $K_n$ is the Monod constant of the consumer cells and $\mu_n$ is the highest rate at which consumer cells can grow when the exchanged molecule is provided in excess. The first term in the constant $\theta$ measures the leakage flux in producing cells ($r_p^l I_p^C$) relative to the flux needed by non-producing cells to grow well ($2\mu_n K_n$). The second term corresponds to the effective uptake rate (active transport with rate $r_n^u$ together with diffusion across the membrane with rate $r_n^l$) in non-producing cells relative to that in producing cells. In the case where cell types have identical rates, $\theta = \frac{r^l I^C}{2\mu K} \equiv \frac{r^l}{\gamma}$ and we thus recover Eq 6.

Moreover, we can show (see S1 Text), that the growth range is given by:

$$R \approx \beta \sqrt{\frac{2(1-\rho)^2 \cdot D}{\rho(2+\rho) \cdot (r_n^u + r_n^l)}} \cdot \ln\left[\delta\left(1 + \sqrt{1 + \frac{4}{\delta}}\right) + 4\right]$$

$$\delta = \frac{r_p^l I_p^C}{2\mu_n K_n} \cdot \frac{2\sqrt{r_p^u + r_p^l}}{\sqrt{r_n^u + r_n^l} + \sqrt{r_p^u + r_p^l}} \cdot \frac{r_n^u + r_n^l}{r_p^u + r_p^l} \tag{8}$$

The first term in the constant $\delta$ measures the leakage flux in producing cells relative to the flux needed by non-producing cells to grow well. The second term corresponds to the diffusion length scale $\propto 1/\sqrt{r_n^u + r_n^l}$ in regions occupied by non-producing cells, relative to the average diffusion length scale. The third term corresponds to the effective uptake rate in non-producing cells relative to that in producing cells. In the case where cell types have identical rates, $\delta = \frac{l^C r_p^l}{2\mu K} \equiv \frac{r^l}{\gamma}$, and we thus recover Eq 4.

## Cellular automaton

In the cellular automaton, cells are placed on a square grid of size 100 x 100 with periodic boundary conditions. Cells interact within an extended Moore neighborhood with range $d_X$, and thus have a neighborhood size of $(2d_X + 1)^2 - 1$. For each cell we calculate the growth rate (using linear growth function) and we randomly pick a cell to reproduce with a probability that is proportional to its growth rate. We then randomly replace one of its neighbors. The grid was initialized with a random arrangement with a given frequency of the two types. We parameterized the experimental community by choosing $d_{\Delta T} = 1$ (8 neighbors) and $d_{\Delta P} = 5$ (120 neighbors, see S2 Table). Simulations were implemented in C++ and in python.

## Experimental communities

The experimental methods are described in detail in reference [11], here we summarize the most relevant details. The community consists of two strains of *Escherichia coli*: ΔT: MG1655 trpC::frt, PR-sfGFP and ΔP: MG1655 proC::frt, PR-mCherry. The deletion of *trpC* and *proC* prevents the production of tryptophan and proline, respectively. Cells are labeled with constitutively expressed fluorescent proteins.

Cells were grown in microfluidic chambers of 60x60x0.76$\mu$m; the small height forces cells to grow in a monolayer. The chambers open on one side into a feeding channel of 22$\mu$m high and 100$\mu$m wide. The microfluidic devices were fabricated from Polydimethylsiloxane (PDMS) using SU8 photoresist molds. Overnight cultures of the two strains (started from single colonies) were mixed in a 1:1 volume ratio and loaded into the microfluidic devices by pipette. Cells were grown on M9 medium (47.8mM Na$_2$HPO$_4$, 22.0 mM KH$_2$PO$_4$, 8.6 mM NaCl and 18.7 mM NH$_4$Cl) supplemented with 1 mM MgSO$_4$, 0.1 mM CaCl$_2$, 0.2% glucose, and 0.1% Tween-20. For the first 10h the medium was supplemented with 434 mM of L-proline and 98 mM of L-tryptophan to allow cells to grow independently and fill the chambers. Subsequently, cells were grown without externally supplied amino acids.

The growth of the communities was followed using time-lapse microscopy. Phase contrast and fluorescent images were taken every 10min using fully automated Olympus IX81 inverted microscopes, equipped with a 100x NA1.3 oil objective, a 1.6x auxiliary magnification, and a Hamamatsu ORCA flash 4.0 v2 sCMOS camera. The sample was maintained at 37°C with a microscope incubator.

## Image analysis

Microscope images were analyzed using Vanellus (version v1.0 [46]). Images were registered, cropped to the area of the growth chambers, and deconvoluted. Images were segmented on the fluorescent channels using custom build Matlab routines [11]. Cell tracking was done using custom build Matlab routines [11] followed by manual correction. Cell length and width were measured by fitting an ellipse to the cell shape and taking the major and minor axis length, respectively. Cell growth rates were estimated by fitting a linear regression to the log transformed cell lengths over a 40min time window [11].

### Data analysis

For each chamber we estimated $P(\Delta T)$ as the number of pixels occupied by $\Delta T$ cells divided by the total number of pixels occupied by cells. We estimated $P(\Delta T|\Delta P, r_{\Delta P})$ by first calculating the local $\Delta T$ frequency for each $\Delta P$ cell, and averaging this over all $\Delta P$ cells within the chamber. The local frequency was calculated as the number of pixels occupied by $\Delta T$ cells divided by the total number of pixels occupied by cells, considering only pixels within $12.1\mu m$ (the interaction range of $\Delta P$) of the cell perimeter. $P(\Delta P|\Delta T, r_{\Delta T})$ was estimated in similar way, using the interaction range of $3.2\mu m$ for $\Delta T$.

To estimate the reduction in community-productivity, we previously developed an in-silico experiment [11]. We used an individual-based biophysical model to calculate the community productivity (measured as the average of the predicted growth rate of all cells) for *clustered communities*, in which the spatial arrangement of cells was based on experimental measurements and compared it to the community productivity in *randomized communities*, in which the spatial arrangement of cells was randomized [11]. For the clustered communities, experimentally observed spatial arrangements of 22 communities were converted to a 40x40 square grid. Each grid site was assigned the cell type that occupied the majority of the corresponding pixels in the real image. Grid sites that remained empty after this procedure were randomly assigned one of the types. This grid was used as input for a previously described individual based model that can predict single cell growth rates [11]. The resulting growth rates were averaged over all cells to obtain the average growth rate for the clustered communities. Subsequently the 40x40 grids were randomized, keeping the frequency of the two cell types constant but permuting their locations. We calculated the average community growth rate for 20 independent randomizations, and we assigned the average over all these permutations as the growth rate of the randomized communities.

## Supporting information

**S1 Text. Local interaction rules for cross-feeding communities.** Here we derive the local interactions rules for 2D and 3D mutualistic communities from the molecular parameters underlying the interaction using a biophysical model. **Fig A: The interaction range is similar for 2D and 3D communities and closely matches the analytical prediction for the growth range**.
(PDF)

**S2 Text. Predicting community level dynamics from local rules using pair-approximation.** Here we derive the community level properties from the local interaction rules by applying pair-approximation to a graph based model.
(PDF)

**S3 Text. Supplementary discussion.** Fig A: Testing assumption regarding replication neighborhood with simulations. Fig B: Validating pair-approximation with cellular automaton simulations. Fig C: Relation between patch size, interaction range, and spatial clustering.
(PDF)

**S1 Table. Parameters values of biophysical model.**
(PDF)

**S2 Table. Parameter values of local interactions rules of experimental cross-feeding community.**
(PDF)

**S3 Table. Robustness of predictions to choice of model parameterization.**
(PDF)

**S1 Video. Simulation of experimental community.** The grid was initialized with a random arrangement and an equal frequency of the two types. The community steady state composition is skewed towards the purple cell type, which represents the proline *E. coli* auxotroph in the experimental community, which has a larger maximum growth rate and a larger interaction range. We parameterized the cellular automaton to represent the experimental community using the parameters shown in S2 Table).
(MP4)

**S2 Video. Simulation of symmetric community.** The two cell types in this simulation have the same interaction range (eight interacting neighbors) and the same maximum growth rate (equal to one). The grid was initialized with a random arrangement and an equal frequency of the two types. At steady state, the community maintains an equal composition of the two types. However, it is possible to see the formation of larger clusters of cells of the same type. As time passes, kin clustering reduces the level of mixing of the two types and thus their growth rates.
(MP4)

**S3 Video. Simulation of asymmetric community.** The two cell types in this simulation have the same interaction range (eight interacting neighbors), but the yellow type has twice the maximum growth rate of the purple type. At steady state, the community composition is skewed towards the type with faster maximum growth rate (more yellow cells). The grid was initialized with a random arrangement and an equal frequency of the two types.
(MP4)

**S4 Video. Simulation of collapsing community.** The two cell types in this simulation have the same interaction range (eight interacting neighbors), but the purple type has a six times faster maximum growth rate compared to the yellow type. The community collapses as predicted by the pair approximation framework, because the maximum growth rates of the two types are very different and the interaction ranges are small.
(MP4)

## Acknowledgments

We thank Michael Doebeli and the members of the Microbial Systems Ecology group, the Doebeli lab, and the Hauert lab for valuable input and discussion.

## Author Contributions

**Conceptualization:** Simon van Vliet, Martin Ackermann, Alma Dal Co.

**Formal analysis:** Simon van Vliet, Christoph Hauert, Alma Dal Co.

**Investigation:** Simon van Vliet, Christoph Hauert, Kyle Fridberg, Alma Dal Co.

**Writing – original draft:** Simon van Vliet, Alma Dal Co.

**Writing – review & editing:** Simon van Vliet, Christoph Hauert, Martin Ackermann, Alma Dal Co.

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
