## [Decision Letter · Decision Letter 0]

2 Aug 2021

Dear Dr. Dal Co,

Thank you very much for submitting your manuscript "Global dynamics of microbial communities emerge from local interaction rules" for consideration at PLOS Computational Biology.

As with all papers reviewed by the journal, your manuscript was reviewed by members of the editorial board and by independent reviewers. In light of the reviews (below this email), we would like to invite the resubmission of a significantly-revised version that takes into account the reviewers' comments.

We cannot make any decision about publication until we have seen the revised manuscript and your response to the reviewers' comments. Your revised manuscript is also likely to be sent to reviewers for further evaluation.

Sincerely,

Jacopo Grilli

Associate Editor

PLOS Computational Biology

Kiran Patil

Deputy Editor

PLOS Computational Biology

Reviewer's Responses to Questions

**Comments to the Authors:**

Reviewer #1: the review is uploaded as a separate file

Reviewer #2: In previous work, the authors examined the growth of two co-dependent species (A and B) that each lack the ability to produce one essential amino acid that is produced by the partner specie. Experiments were conducted in 2D layers. They found that a cell could benefit from partner specie cells only if these where within a small interaction range set by the uptake rate of the molecule. They also found that the growth rate of a cell depended on the fraction of cells of the partner species within its interaction range and that maximum growth rate was set by the leakage rate of the exchanged amino acid.

In this paper the authors use a graph-base model to show that these local rules of interaction are sufficient to predict several global community measures such as community composition, spatial arrangement, and growth rate.

The main findings are:

- Developed a simple graph-based model that encapsulates local rules of interactions and measurements of microscopic parameters and that can predict community-level properties. Improvement on past graph-based models is that the interaction range can be different for different cells/metabolites, which the author show is necessary to understand the experimental community.

- Analytical expressions for the steady state global and local compositions of the community as a function of the local rules, including predictions for differences between 2D and 3D communities.

- Exploration of the effects of different neighborhood sizes, from very short range to well-mixed, and different growth rates, and prediction of the range of parameter values that leads to the collapse of the community.

- Validation with experimental measurements of microbial community.

These are all good results that will be useful for the field. Being able to make predictions that cross scales is rare and extremely valuable. However, in its current form the paper risks being ignored because it is poorly written and organized, which makes it extremely tedious to read.

Main Concerns:

The abstract very long and unfortunately vague. It fails to clearly spell out the key new finding as opposed to what was learned from the previous study. This is especially true for lines 25-30: “For these cross-feeding communities, the community growth rate is reduced when cells interact only with few neighbors; as a result, some communities can co-exist in a well-mixed system, where cells can interact with all other cells, but not in systems where cells can interact only with close by neighbors.” What is new and what is past results in that statement? I recommend rewriting the abstract entirely, making sure to clearly spell out the key new findings.

The artificial split between the Model section and the Result section forced me to go back and forth between the two section and got me lost several times. This is a Computational Biology paper. The model is one of the main contributions of the paper. It is tedious to go through the model first without discussion of the plots which provide meaning to the equations. I recommend reorganizing so that derivations of key equations and plots and interpretation are together. For example: the sections “Steady state community properties” and “Local cell-cell interactions predict community properties” and Fig 3 clearly belong together.

This lack of organization reflects the poor overall organization of the paper. In its current form it is difficult for the reader to follow the logic of question – model – plot – interpretation, next question – model – plot – interpretation, and so on… instead the same question/concept is treated in various places. For example, the results on community collapse are all over the places: could all text/figures/equations related to the analysis of collapse in cross feeding communities be in one place?

Minor Comments:

SI line 249: you set r_A <= r_B and the assume that I_A, the set of all cells from which a given cell A receives metabolites, is included or equal to I_B, the set of all cells from which a given cell B receives metabolites. Can you explain/justify/demonstrate how r_A <= r_B implies that I_A is included or equal I_B? In the current text there is no explanation about it, or maybe it is distributed in various places.

Fig 3 caption: “(B) The neighborhood size has only a minor effect on the global composition of the the community”. I am not sure I agree with this wording. Even asymptotically there seem to be deviations from the well-mixed system on the order of ~20%? (green lines). Note also the repeated word.

**Have the authors made all data and (if applicable) computational code underlying the findings in their manuscript fully available?**

Reviewer #1: Yes

Reviewer #2: None

PLOS authors have the option to publish the peer review history of their article (what does this mean?). If published, this will include your full peer review and any attached files.

Reviewer #1: No

Reviewer #2: No
---

## [Decision Letter · Decision Letter 1]

28 Jan 2022

Dear Dr. Dal Co,

We are pleased to inform you that your manuscript 'Global dynamics of microbial communities emerge from local interaction rules' has been provisionally accepted for publication in PLOS Computational Biology.

Best regards,

Jacopo Grilli

Associate Editor

PLOS Computational Biology

Kiran Patil

Deputy Editor

PLOS Computational Biology

Reviewer's Responses to Questions

**Comments to the Authors:**

Reviewer #1: The ample rewriting of the manuscript improved its clarity and makes it an even more pleasant read.

I am overall satisfied with the answers the authors provided to my comments and the consequent modifications.

I only have a couple of minor further observations.

First, regarding the statement (line 479) that the reduction in community productivity is significant. I understand that the authors cannot estimate the error in the theoretical estimate based on published parameters. However, they can asses whether their experimentally measured frequencies are significantly different from the frequency predicted in the well-mixed case and from any of the two predicted values when interactions are local (a t-test should do). Ideally, they should be significantly different from the former and not from the latter.

I would also add the confidence interval to the value 0.19 on line 400, instead of just in the figure caption.

Second, I was initially confused by the fact that maximum growth rate could be different for the two strains of this experimental system. In principle, I would imagine that, given infinite amount of the amino acid they do not produce, the two strains should grow similarly. But I then realized that what the authors call maximum growth rate is the rate of growth of a cell of one type whose finite neighbourhood is filled with cells of the other type. This thus depend on how much vital amino acid the partner is able to produce.

I think this point could be made more explicit (for instance around line 377), given the importance of growth rate differences in the model.

A small technical point that I am not sure has any relevance. Upon printing, only data in panels C and D of figure 6 were reproduced. The figure displayed instead perfectly as a separate panel and on the screen.

Reviewer #2: The authors have addressed my comments.

**Have the authors made all data and (if applicable) computational code underlying the findings in their manuscript fully available?**

Reviewer #1: Yes

Reviewer #2: Yes

PLOS authors have the option to publish the peer review history of their article (what does this mean?). If published, this will include your full peer review and any attached files.

Reviewer #1: No

Reviewer #2: No

---

## [Editor Report · Acceptance letter]

1 Mar 2022

PCOMPBIOL-D-21-01105R1 

Global dynamics of microbial communities emerge from local interaction rules

Dear Dr Dal Co,

I am pleased to inform you that your manuscript has been formally accepted for publication in PLOS Computational Biology. Your manuscript is now with our production department and you will be notified of the publication date in due course.

With kind regards,

Orsolya Voros
